# The P2X7 Receptor Promotes Colorectal Inflammation and Tumorigenesis by Modulating Gut Microbiota and the Inflammasome

**DOI:** 10.3390/ijms23094616

**Published:** 2022-04-21

**Authors:** Claudio Bernardazzi, Morgana Teixeira Lima Castelo-Branco, Beatriz Pêgo, Beatriz Elias Ribeiro, Siane Lopes Bittencourt Rosas, Patrícia Teixeira Santana, João Carlos Machado, Camille Leal, Fabiano Thompson, Robson Coutinho-Silva, Heitor Siffert Pereira de Souza

**Affiliations:** 1Department of Pediatrics, University of Arizona, Tucson, AZ 85724, USA; claudiobma1@gmail.com; 2Department of Clinical Medicine, Federal University of Rio de Janeiro, Rio de Janeiro 21941-913, Brazil; morgcb@gmail.com (M.T.L.C.-B.); biapdamasceno@gmail.com (B.P.); bakerribeiro@gmail.com (B.E.R.); sianeros@gmail.com (S.L.B.R.); pattsant@gmail.com (P.T.S.); 3Institute of Biomedical Sciences, Federal University of Rio de Janeiro, Rio de Janeiro 21941-590, Brazil; 4Biomedical Engineering Program, COPPE, Federal University of Rio de Janeiro, Rio de Janeiro 21941-901, Brazil; jcm@peb.ufrj.br; 5Institute of Biology, Federal University of Rio de Janeiro, Rio de Janeiro 21941-599, Brazil; camille.victoria@gmail.com (C.L.); fabianothompson1@gmail.com (F.T.); 6Institute of Biophysics Carlos Chagas Filho, Federal University of Rio de Janeiro, Rio de Janeiro 21941-590, Brazil; rcsilva@biof.ufrj.br; 7D’Or Institute for Research and Education (IDOR), Rua Diniz Cordeiro 30, Botafogo, Rio de Janeiro 22281-100, Brazil

**Keywords:** colitis-associated colorectal cancer, gut microbiota, inflammatory bowel disease, P2X7R, purinergic signaling

## Abstract

**Background:** Given the role of the P2X7 receptor (P2X7R) in inflammatory bowel diseases (IBD), we investigated its role in the development and progression of colitis-associated colorectal cancer (CA-CRC). **Methods:** CA-CRC was induced in P2X7R^+/+^ and P2X7R^−/−^ mice with azoxymethane (AOM) combined with dextran sodium sulfate (DSS). In a therapeutic protocol, P2X7R^+/+^ mice were treated with a P2X7R-selective inhibitor (A740003). Mice were evaluated with follow-up video endoscopy with endoluminal ultrasound biomicroscopy. Colon tissue was analyzed for histological changes, densities of immune cells, expression of transcription factors, cytokines, genes, DNA methylation, and microbiome composition of fecal samples by sequencing for 16S rRNA. **Results:** The P2X7R^+/+^ mice displayed more ulcers, tumors, and greater wall thickness, than the P2X7R^−/−^ and the P2X7R^+/+^ mice treated with A740003. The P2X7R^+/+^ mice showed increased accumulation of immune cells, production of proinflammatory cytokines, activation of intracellular signaling pathways, and upregulation of *NLRP3* and *NLRP12* genes, stabilized after the P2X7R-blockade. Microbial changes were observed in the P2X7R^−/−^ and P2X7R^+/+^-induced mice, partially reversed by the A740003 treatment. **Conclusions:** Regulatory mechanisms activated downstream of the P2X7R in combination with signals from a dysbiotic microbiota result in the activation of intracellular signaling pathways and the inflammasome, amplifying the inflammatory response and promoting CA-CRC development.

## 1. Introduction

One of the most feared complications resulting from chronic colonic inflammation in ulcerative colitis (UC) and Crohn’s disease (CD), collectively known as inflammatory bowel disease (IBD), is the development of colitis-associated colorectal cancer (CA-CRC). Although improvements in endoscopic surveillance protocols have allowed the earlier detection of dysplasia and subsequent CA-CRC [1], the diagnosis and management of dysplasia is still a major challenge in patients with IBD [2]. In addition to the increased risk in patients with UC [3] and with colonic CD [4], some studies have indicated that patients with CA-CRC have inferior survival compared to patients with sporadic CRC [5].

While the development of sporadic CRC is well established and characterized by histological and genetic changes known as the adenoma-carcinoma sequence, CA-CRC develops through a different sequence. Chronic inflammation of IBD induces mutations and creates a regenerative background resulting in a distinctive selective pressure that may lead to CA-CRC [6]. Although mutations leading to CRC are similar in IBD and in the sporadic type, the order by which they occur appears to be different [7]. These observations indicate the need to elucidate the pathogenesis of CA-CRC, including the molecular and immunologic mechanisms underlying the chronic inflammatory process of IBD, to guide the development of new therapies to reach deep remission and lower the risk of CA-CRC.

Extracellular adenosine triphosphate (ATP) released from tissue injury resulting from infection or cellular stress is regarded as a damage-associated molecular pattern (DAMP) capable of regulating several physiological cell functions through the interaction with the purinoreceptor P2X7 (P2X7R), an extracellular ATP-gated cation channel expressed on both epithelial and immune effector cells [8]. The modulatory effect of the ATP-P2X7R pathway in T regulatory cell function [9], the production of proinflammatory cytokines [10], the regulation of cell death [11], the elimination of infectious pathogens [12,13,14], and inflammasome activation [15,16] indicates its role in the pathogenesis of autoimmune diseases [17,18] and chronic inflammatory disorders such as IBD [19].

Given the already mentioned information regarding the ATP-P2X7R pathway and our previous findings demonstrating site-specific modulation of P2X7 in intestinal epithelial cells [11], we hypothesized that purinergic signaling through P2X7R plays a critical role in the regulation of chronic intestinal inflammation and consequently may be involved in the pathogenesis of CA-CRC. Therefore, in the present study, we investigated the potential therapeutic effect of P2X7R blockade in an experimental model of CA-CRC.

## 2. Results

### 2.1. P2X7R Blockade Attenuated Inflammation and Tumorigenesis and Reduced Morbidity and Mortality in the Mouse Model of AOM/DSS-Induced CA-CRC

#### Video Colonoscopy Associated with eUBM

To follow up on the development of CA-CRC in vivo, we periodically examined the mice by video colonoscopy during the intervals of DSS administration. As in previous works from our group [20], colonoscopy was associated with eUBM, a device that utilizes a high-frequency ultrasound system that allows the acquisition of detailed cross-sectional images (Appendix A). After the first cycle of DSS (3rd week), most animals presented mucosal edema and an abnormal vascular pattern. Ultrasonic images revealed a practically normal wall thickness and a clear distinction between the hyperechoic and hypoechoic layers. After the second cycle of DSS (6th week), all P2X7R^+/+^ wild-type mice displayed mucosal granularity and friability, and most of them also had ulcers and bleeding. The P2X7R^+/+^ wild-type mice treated with A740003 did not present the entire range of abnormalities observed in the untreated animals. However, in some cases, the colonic mucosa showed friability, and edema was also corroborated by the ultrasonic images showing focal increases in the wall thickness. Similar changes were observed among the AOM/DSS-induced P2X7R^−/−^ animals, while no change was detected among the P2X7R^−/−^ control animals. After the third cycle of DSS (week 9), the colonic abnormalities were strongly aggravated in the AOM/DSS-treated P2X7R^+/+^ animals. The colonic mucosa showed advanced inflammatory changes but also the formation of polyps. The ultrasonic images revealed wall thickening of the hyper- and hypoechoic layers, lacking a clear distinction among each other. Some of the induced wild-type mice treated with A740003 showed mucosal edema, vascular changes, and discrete elevations in the colonic mucosa without clear inflammatory abnormalities; however, the well-defined layers under the ultrasonic images were preserved, indicating superficial changes only. Most P2X7R^−/−^ mice showed a normal colon during the whole experiment, but in some cases, mild inflammatory changes could be detected at the end of the protocol; however, there were no ulcers or polyps (Figure 1).

### 2.2. Survival, Body Weight, Colon Length, and Number of Tumors

As shown in Figure 1C, the AOM/DSS-induced P2X7R^+/+^ mice treated with A740003 had a longer survival than the AOM/DSS-induced P2X7R^+/+^ mice during the experimental period. Body weight significantly decreased at the end of week 4 (Day 28) in the AOM/DSS-induced P2X7R^+/+^ mice compared to the control P2X7R^+/+^ mice. Upon treatment termination at the end of week 8 (Day 56), body weight changed even more in the AOM/DSS-induced P2X7R^+/+^ mice than in the A740003-treated and control P2X7R^+/+^ mice (Figure 1D). The colon length shortened more in the AOM/DSS-induced P2X7R^+/+^ mouse group than in the control group and in the A740003-treated mouse group (Figure 1E,F). Furthermore, the number of polyps/tumors was significantly greater in the AOM/DSS-treated P2X7R^+/+^ mice than in the A740003-treated mice. The control P2X7R^+/+^ mice and the P2X7R^−^^/^^−^ mice with or without AOM/DSS induction presented no tumors (Figure 1G). Polypoid tumors were found in the middle and distal colon in the AOM/DSS-treated mice (80% of the animals); however, A740003 treatment significantly prevented the development of tumors in the AOM/DSS-treated mice (33% of the animals). Multiple tumors occurred in 41.6% of the AOM/DSS-induced P2X7R^+/+^ mice, whereas the A740003-treated group had only single tumors, whenever present. Upon histological evaluation, we found no metastatic tumors in other tissues, such as the liver or spleen, of the AOM/DSS-treated mice (data not shown). In the AOM/DSS-treated mice, 90% of the lesions were adenomas, and 10% were adenocarcinomas. In contrast, in the A740003-treated animals, no adenocarcinomas were detected, with increases in lesions that were either hyperplastic polyps (50%) or adenomas (50%).

### 2.3. P2X7R Blockade Attenuated Colonic Injury in the AOM/DSS-Treated Mice

#### Histopathologic Assessment of Tissue Samples

The histological assessment confirmed significantly increased inflammation and the development of hyperplastic changes, including tumors, in the AOM/DSS-induced P2X7R^+/+^ mice compared with the control mice and with the AOM/DSS-induced mice treated with A740003. However, no tumor formation and practically no inflammatory response were observed in the AOM/DSS-induced P2X7R^−^^/^^−^ mice (Figure 2). These data suggest that the P2X7R^−^^/^^−^ mice and the P2X7R^+/+^ mice treated with the P2X7R antagonist (A740003) develop less inflammation and consequently less hyperplasia and tumors following AOM/DSS induction.

Analysis of the density of collagen fibers in the colon tissue revealed enhancement of fibrosis in the AOM/DSS-induced P2X7R^+/+^ mice compared with the control mice and with the AOM/DSS-induced mice treated with A740003. In the epithelial compartment of the colon, the density of mucous-secreting goblet cells was significantly lower in the AOM/DSS-induced P2X7R^+/+^ mouse samples than in the samples from the normal control group or the AOM/DSS-induced mice treated with A740003. No significant changes were detected between the P2X7^−^^/^^−^ groups (Appendix A).

### 2.4. Regulation of Cell Proliferation and Apoptosis in the Colon

To determine the role of cell proliferation and cell death in this model, we stained cells with an anti-Ki67 antibody by immunohistochemistry and used a TUNEL assay to label apoptotic cells. Colon samples from nondysplastic inflamed areas of the AOM/DSS-induced P2X7R^+/+^ mice showed significantly lower rates of Ki67-positive cells than those of the control group. Regarding apoptosis, colon samples from nondysplastic inflamed areas of the AOM/DSS-induced P2X7R^+/+^ mice showed significantly higher rates than those of the control group and the AOM/DSS-induced mice treated with A740003. In contrast to the inflamed areas, the analysis of samples from tumor areas revealed an opposite result, with less apoptosis and more Ki67-positive proliferating cells, as expected (Figure 3).

### 2.5. P2X7R Blockade Attenuated the Accumulation of Immune Cells and the Production of Inflammatory Cytokines in the Colon

#### 2.5.1. Accumulation of Immune Cells in the Colon

To characterize the different cell populations present in the colonic lamina propria, we labeled CD4 and CD11b cells by immunohistochemistry. The inflammatory cell infiltrates observed within the lamina propria of nondysplastic inflamed areas of the AOM/DSS-induced P2X7R^+/+^ mice showed an increased concentration of both CD4- and CD11b-positive cells. Treatment with A740003 significantly attenuated the accumulation of CD4- and CD11b-positive cells in the AOM/DSS-induced mice. In contrast to the inflamed areas, the analysis of samples from tumor areas revealed sparsely distributed CD4- and CD11b-positive cells (Appendix A).

#### 2.5.2. Production of Inflammatory Cytokines in the Colon

Next, we investigated whether P2X7R could affect the production of inflammatory mediators potentially involved in the inflammatory and hyperplastic changes associated with AOM/DSS induction. For this purpose, we used a mouse Th1/Th2/Th17 cytokine kit based on bead array technology to simultaneously detect IFN-gamma, TNF, IL-2, IL-4, IL-6, IL-10, and IL-17A. The analysis of the supernatants obtained from colon explant cultures revealed that the concentrations of TNF-alpha, IL-17A, and IL-6 were significantly increased in the samples from the P2X7R^+/+^ AOM/DSS-treated mice compared to the samples from the controls and the A74003-treated mice. However, the concentration of IL-10 was significantly lower in the P2X7R^+/+^ AOM/DSS-treated mice than in the control mice. The P2X7R^−^^/^^−^ samples did not show any significant changes. Only the quantifiable results were analyzed (Figure 4).

### 2.6. P2X7R Blockade Modulated the Expression of Genes Related to Inflammation and Cancer in the Colon

To investigate the possible mechanisms by which AOM/DSS might regulate the inflammatory response and dysplastic changes in the colon, we examined the mRNA expression of genes potentially involved in tissue remodeling and inflammatory and protumorigenic processes. The results showed an overall tendency for upregulation of the expression of all target genes studied in the P2X7^+/+^ AOM/DSS-induced mice. However, significant changes compared to the controls and the A74003-treated mice were observed only for the *NLRP3*, *NLRP12*, *MAPK14*, and *IL-1beta* genes (Figure 5). Other genes analyzed in this study, including *AKT-1*, *BCL-2*, *TP-53*, *CXCR3*, *STAT-3*, and *CASP-1,* did not show significant changes among the experimental groups (Appendix A).

### 2.7. P2X7R Blockade Reduced the Activation of Intracellular Pathways in the Colon

To reinforce the findings obtained with qPCR, we also investigated the NF-kappa B and phosphorylated ERK MAP kinase intracellular signaling pathways at the protein level by immunohistochemistry. NF-kappa B and p-ERK displayed similar expression patterns and tissue distributions. NF-kappa B and p-ERK were present in the epithelium and the lamina propria mononuclear cells at significantly higher densities in the nondysplastic inflamed areas of the P2X7^+/+^ AOM/DSS-induced mice than those of the controls and the A740003-treated mice. However, the tumor areas showed only a limited number of NF-kappa B- and p-ERK-positive cells (Figure 6).

### 2.8. The Expression of P2X7R and Beta-Catenin in the Colon

Wnt/beta-catenin signaling, another critical pathway in the development of CRC, was analyzed by immunohistochemistry. Beta-catenin was detected among lamina propria mononuclear cells and in the epithelial compartment, particularly toward the bottom of the crypts in the nondysplastic inflamed colon, compared to strong staining of the dysplastic epithelium in tumors. Although the semiquantitative analysis showed that the overall expression of beta-catenin did not differ significantly between the groups, they were clearly qualitatively distinct. While beta-catenin distribution showed a diffuse pattern in the nondysplastic inflamed colon, in the tumor, in addition to cytosolic localization, strong nuclear staining was observed in the dysplastic epithelium. However, parallel staining with anti-P2X7R antibody revealed significantly higher expression in the lamina propria and the epithelial compartment of nondysplastic inflamed tissue than in the epithelium of the tumors, which showed faint coloration (Appendix A).

### 2.9. P2X7R Blockade Changed the Methylation Patterns of Specific Genes in the Colon

To evaluate the DNA methylation status of CRC-related genes in the experimental groups, we assessed the promoter regions of *p53*, *p16*, *MLH1*, *Gja9*, *Igfbp3,* and *APC* in a semiquantitative fashion. Of all target genes studied, we identified differential methylation only in the *Gja9* gene, which was more pronounced in the P2X7^+/+^ AOM/DSS-induced mice (Appendix A).

### 2.10. P2X7R Blockade Modulated the Microbiota Associated with Inflammation and Tumors in the Colon

High-throughput sequencing produced 4,335,464 16S rRNA sequences (185 nt length) after Deblur quality control clean-up from 36 samples (P2X7R^+/+^ control 8797 ± 1889; P2X7R^+/+^ AOM/DSS 110,042 ± 15,288; A740003-treated P2X7R^+/+^ AOM/DSS 122,288 ± 26,596; P2X7R^−^^/^^−^ control 166,007 ± 52,738; P2X7R^−/−^ AOM/DSS 173,031 ± 68,893; values represent the mean ± SD). The reads were delineated into 493 operational taxonomy units (OTUs). Estimators of alpha diversity were calculated according to the Shannon index. Although no significant difference was detected among the groups, we observed a trend toward a decreased diversity in the AOM/DSS-induced P2X7R^+/+^ group (Figure 7A). Principal coordinate analysis (PCoA) revealed a clear clustering of P2X7R^−^^/^^−^ (triangles) away from P2X7R^+/+^ samples (circles). Among the P2X7R^+/+^ samples, unique structural changes in the fecal microbiota appear to have occurred after P2X7R blockade with A740003, as most samples shifted away from AOM/DSS-induced P2X7R^+/+^ (highlighted with colored oval forms). However, the P2X7R^+/+^ controls were distributed unevenly, revealing the heterogeneity within that group, which might have prevented a more comprehensive and accurate analysis. In summary, we found a dissimilarity in the fecal microbiota between the A740003-treated and untreated AOM/DSS-induced P2X7R^+/+^ mice and between the P2X7R^+/+^ and P2X7^−^^/^^−^ controls (Figure 7B). The Venn diagram showed that only a minority of OTUs were preserved and shared among groups, with a greater superimposition observed among the AOM/DSS-induced P2X7R^+/+^ (53%) and P2X7^−^^/^^−^ (44%) groups (Figure 7C). As expected, the fecal microbiota was mainly composed of Bacteroidetes and Firmicutes in all groups, followed by less abundant Proteobacteria and Epsilonbacteraeota, among others (Appendix A). At the phylum level, compared to the P2X7^+/+^ control group, the AOM/DSS-induced P2X7R^+/+^ group presented a trend toward a higher relative abundance of Firmicutes (4–20% vs. 13–31%) and Tenericutes (0–0.4% vs. 0.1–1.4%), whereas Fusobacterium was exclusively identified in the AOM/DSS-induced P2X7R^+/+^ group (Figure 7D). Furthermore, we observed that bacteria belonging to the *Mycoplasma* and *Mucispirillum* genera (of the Tenericutes and Deferribacteres phyla, respectively) had a higher relative abundance in the AOM/DSS-induced P2X7R^+/+^ group than in the P2X7^+/+^ control group. In contrast, compared to the P2X7R^+/+^ control group, the P2X7R^−^^/^^−^ group presented a trend toward a higher relative abundance of Cyanobacteria (0–1.9% vs. 0.3–4.3%) and Spirochaetes (0–0 vs. 0.1–0.4) (Figure 7D, and Appendix A). The Firmicutes:Bacteroidetes ratio was not significantly different among the groups (Appendix A).

## 3. Discussion

In this study, we investigated whether purinergic signaling through P2X7R could play a role in the development of tumors in a murine model of CA-CRC. The results showed that chemical P2X7R blockade markedly abrogated the development of tumors, whereas the P2X7R^−^^/^^−^ mice did not develop CA-CRC. In particular, we demonstrated that prophylactic treatment with the P2X7R antagonist stabilized mucosal cell turnover and substantially attenuated morphologic and molecular abnormalities underlying the chronic inflammatory process of the colon. Moreover, we detected subtle changes in intestinal microbial components and the methylation patterns of genes related to CRC.

Current data regarding the role of P2X7R in human CRC derive basically from descriptive studies associating the increased expression of P2X7R in tumors with the poorer survival and prognosis of patients [21,22,23]. Previous studies investigating the potential role of the P2X7R in tumorigenesis, provided contradictory results. However, only a few studies have investigated the role of the P2X7R in CRC. For example, in an animal model of tumor growth using B16 melanoma and CT26 colon carcinoma cells, the absence of the P2X7R correlated with increased growth, metastasis, and an inefficient antitumor immune response [24]. In another study, anticancer chemotherapy was shown to be less responsive against tumors established in P2X7R-deficient animals [25]. On the other hand, Sougiannis et al., recently showed that emodin, an anthraquinone with antitumorigenic properties, was effective against the onset of genetic and AOM/DSS-induced CRC, via the reduction of M2-like protumorigenic macrophages [26]. These data appear to corroborate the results from a previous study demonstrating that emodin could inhibit carcinogenesis-associated intestinal inflammation, preventing the development of CA-CRC in an AOM/DSS model [27].

The effect of P2X7R activation by ATP in CRC in vitro with cancer cell lines and in vivo using xenograft tumor models has been investigated recently. In one of the studies, the investigators concluded that ATP-P2X7R activation promotes the migration and invasion of CRC cells, possibly through the activation of the STAT3 pathway [28]. In another study, Zhang et al. demonstrated that A438079, another selective PX7R antagonist, inhibited cell proliferation, migration, and invasion, but promoted cell death in the CRC cell lines, HCT-116 and SW620, and in SW620 cell xenografted BALB/c nude mice [29]. Taken together, these data appear to support our findings on the influence of P2X7R in the development of CRC. However, we utilized a model based on a chronic inflammatory background to mimic human CA-CRC. In fact, upon tissue collection on week eight, after the third cycle of DSS, we continued to detect inflammation and demonstrated that most components involved in the chronic inflammatory process, including the overexpression of P2X7R, were predominantly located in the nondysplastic surroundings of the tumors. This finding appears to be consistent with the idea that high concentrations of ATP, within the inflammatory process and in close vicinity of inflammation, open large pores that release inflammatory mediators and can result in apoptotic cell death. In contrast, low concentrations of ATP, distant from inflammatory hotspots, open cation channels that may result in cell proliferation [30].

However, in contrast to our findings, in a previous study using the AOM/DSS model including P2X7R^−^^/^^−^ mice and treating animals with the selective PX7R antagonists A438079 and A740003, the investigators achieved opposing results. More tumors were detected among the P2X7R^−^^/^^−^ mice and the animals treated with P2X7R antagonists, including a large number of adenocarcinomas [31]. It is possible that differences in the protocols, such as therapeutic versus prophylactic intervention, and animal housing and care, for example, may have influenced the different outcomes. However, the diametrical opposing results of the two studies led us to search for another more reasonable explanation. In fact, our analysis of the intestinal microbiota appears to have shed important light on the critical role of microbial signaling and its presence in the inflammatory process and in tumorigenesis in the CA-CRC model. Hence, it is likely that differential profiles of the intestinal microbiota in the experimental groups constituting the two studies might contribute to the distinct outcomes but render data difficult to compare directly. Moreover, these discrepant results may indicate the likely need for previous conditioning of animals and possibly also analyzing the microbiota before experimental induction.

Similar to previous publications using the AOM/DSS model [32,33], this study showed that animals developing tumors invariably displayed inflammatory changes in the colon and lost more weight, had shorter colon lengths, and showed overall reduced survival. In particular, the endoscopic follow-up carried out in this study showed that most tumors developed after the third cycle of DSS in the P2X7R^+/+^, but not in the P2X7R^−^^/^^−^ mice, and at significantly lower rates in the P2X7R^+/+^-induced mice treated with the selective blockade of P2X7R. Similar to previous studies reporting the effects of P2X7R activation in the intestine, the nondysplastic areas of the colon of the AOM/DSS-induced P2X7R^+/+^ animals displayed characteristic chronic inflammatory changes combined with epithelial cell loss [34], including mucous-producing cells, increased fibrogenesis [35], and accumulation of mononuclear cells, with increased production of Th1 and Th17 proinflammatory cytokines [36,37], decreased production of IL-10 [9], and upregulation of IL-1 beta expression [19,31]. A prolonged imbalance of cytokines and chemokines within the intestinal mucosa is thought to favor CA-CRC development by various mechanisms affecting epithelial cells. For example, signals from TNF-alpha, IL-17, and IL-6 appear to foster a tumor-supportive microenvironment, probably via mitogenic changes [38], which can impact epithelial cell migration and survival programs [39]. In addition, TNF-alpha, IL-17, and IL-6 have also been shown to induce tumor growth through NF-kappa B and STAT3 activation [40,41].

Regarding the activation of intracellular signaling pathways, this study investigated several molecules usually involved in tumorigenesis, including CRC development, such as STAT3, p53, Akt1, and Bcl2. Nevertheless, the analysis showed that the levels of NF-kappa B and mitogen-activated protein kinases (MAPK) are upregulated in the colon of the P2X7R^+/+^ AOM/DSS-induced mice. NF-kappa B represents a well-established proinflammatory and protumor pathway both in humans and in experimental IBD [42,43] and was increased in the intestinal mucosa of the mice developing tumors in this study. In the AOM/DSS mouse model, the canonical NF-kappa B pathway was previously shown to promote tumorigenesis through the stimulation of myeloid cells, mediating the production of IL-1-beta and proinflammatory cytokines, including TNF-alpha, and acting on epithelial cells by inhibiting apoptosis [44]. In addition, NF-kappa B signaling enhanced Wnt activation and promoted dedifferentiation of non-stem cells, which developed a tumorigenic capacity [45]. Concerning MAP kinases, we demonstrated that the colon of the P2X7R^+/+^ AOM/DSS-induced mice shows upregulation of MAPK14 expression and overexpression of extracellular signal-regulated kinase (ERK). The upregulation of both MAPK14 and ERK expression was blocked in the P2X7R^−^^/−^ mice and in the mice treated with the P2X7R antagonist. ERK is a downstream protein belonging to the Ras-Raf-MEK-MAPK/ERK signaling pathway and commonly shows upregulated expression in tumorigenesis [46,47]. Similar to our results, previous data have shown that the AOM/DSS experimental model enhances the production of ERK, which is involved in cell proliferation [48]. Characteristically, NF-kappa B and phosphorylated-ERK were strongly expressed in lamina propria mononuclear cells in inflamed areas and epithelial cells of the tumors. Our data suggest that purinergic signaling through P2X7R may modulate the activation of intracellular signaling pathways involved in the inflammatory and tumorigenic response in the intestinal mucosa.

Although the participation of the inflammasome in the pathogenesis of IBD and CRC has been investigated in recent years, its intricate mechanisms have resulted in some contradictory results in both humans and experimental models [49]. Regarding NLR molecules involved in the inflammasome activation investigated here, NLRP6 and NLRC4 mRNA levels did not differ among the experimental groups, while upregulated NLRP12 expression paralleled that of NLRP3 in the P2X7R^+/+^ AOM/DSS-induced mice. Interestingly, the cytosolic recognition of microbial components resulting in NLRP3 activation was promoted by pannexin-1 in the presence of ATP [50]. Taken together, these findings strongly suggest that P2X7R associated with pannexin-1 and the consequent downstream NLRP3 activation are implicated in the pathogenesis of AOM/DSS-mediated inflammation and tumor development. Although dysfunction of NLRP12 has been reported in some inflammatory disorders [51], in vitro studies demonstrated that NLRP12 could inhibit canonical and noncanonical NF-kappa B pathways [52]. Moreover, mice deficient in NLRP12 were more susceptible to colitis and colorectal tumorigenesis [53]. Although caspase-1 expression was not significantly upregulated in the current study, IL-1 beta, a downstream molecule critically involved in inflammasome activation, showed strongly upregulated expression. Nonetheless, the differential upregulation of NLR expression observed at the end of the protocol may not reflect the dynamic changes that might have occurred during the experiment and may represent a limitation of this study.

The mechanism by which DSS induces colitis in mice is still poorly understood; however, studies have identified several factors, including changes in the intestinal microbiota [54], an effect partially prevented by treatment with antibiotics [55]. In particular, DSS-induced colitis has been shown to favor an increase in the abundance of *Enterobacteriaceae*, *Bacteroidaceae,* and *Clostridium spp*. [56]. In contrast to our results, most AOM/DSS model investigations identified an increased relative abundance of Proteobacteria, Deferribacteres, Verrucomicrobia, and *Bacteroides* and a decreased abundance of *Prevotella* [57]. In a previous study, the chemical blockade of P2X7R reversed the taxonomic alterations of Bacteroidetes and Verrucomicrobia as well as *Akkermansia* in ethanol-fed mice [58]. This finding appears to be in accordance with our results, at least in part, showing dissimilarity of gut microbiota between the A740003-treated and untreated AOM/DSS-induced P2X7R^+/+^ mice and between the P2X7R^+/+^ and P2X7^−/^^−^ controls. In addition, bacteria belonging to the *Mycoplasma* and *Mucispirillum* genera had a higher relative abundance, whereas Fusobacterium was exclusively identified in the AOM/DSS-induced P2X7R^+/+^ group. Similar to our results, the phylum Tenericutes was positively associated with CA-CRC incidence and tumor burden in the AOM/DSS experimental model [55]. Among the Tenericutes, the genus *Mycoplasma* has also been recently associated with CRC, suggesting a potential role in colonic tumorigenesis [59]. Corroborating our findings, *Mucispirillum* has also been associated with the development of CA-CRC [60]. Here, we show for the first time, to the best of our knowledge, a trend toward a higher relative abundance of Cyanobacteria and Spirochaetes among the P2X7R^−^^/^^−^ samples, suggesting a possible protective role against the development of CA-CRC. However, the lack of statistical significance and the only partial restoration with the chemical antagonist of P2X7R suggest that bacterial changes observed in this study may represent a secondary phenomenon and are not tumorigenic per se in isolation. In fact, the results appear to reinforce the fundamental role of the ATP/P2X7R pathway in the underlying context of tumor development in this model.

Selective hypermethylation of the promoter regions of tumor suppressor genes with hypomethylation of the tumor genome has been previously reported as a common epigenetic modification in CRC. Moreover, mutations involved in the pathogenesis of CRC are usually regarded as similar to those underlying CA-CRC, but the order in which they arise is rather different [7]. In this study, our results involving *p53, p16, MLH1, Igfbp3,* and *APC* were less consistent than those of *Gja9* and did not allow clear differentiation among the experimental groups. In turn, *Gja9* was shown to be relatively hypermethylated in the AOM/DSS-induced P2X7R^+/+^ mice. *Gja9* belongs to the connexin family of proteins that constitute gap junctions, often compromised in cancers, including CRC [61]. However, the overexpression of beta-catenin shown here is compatible with the aberrant Wnt/beta-catenin signaling pathway possibly triggered by the loss of *APC* function [62]. The relatively low expression of P2X7R in the dysplastic area, where beta-catenin expression is high, with a strong presence in the nuclei, suggests that P2X7R may not participate directly in the growth of the dysplastic epithelium. In addition, the neighboring inflammation with an active ATP/P2X7R pathway might have fueled tumor growth with the additional or alternative participation of the activated NF kappa-B pathway [63]. Taken together, these data highlight the complexity of tumor development in the context of colitis, particularly amplified by the dynamic nature of interactions and epigenetic modifications during the process. Therefore, it is likely that data regarding gene methylation in this study should be interpreted with caution, not only due to the semiquantitative nature of the experiments but also because our protocol was restricted to a transversal analysis at the end of the protocol when the most critical interactions and transformations might have occurred, and cumulative effects cannot be distinguished.

Although this study presents a successful approach for the development of CA-CRC and its association with the ATP/P2X7R pathway, important limitations should be acknowledged. First, the number of experiments and animals per experiment was relatively small, mainly owing to technical difficulties regarding the care of animals and the long-term follow-up. However, the overall results were broadly consistent and showed several significant differences. Therefore, the successful data generated in this study should serve, at least in part, as a pilot for further investigations. Future studies on the subject should also consider other dosages and other P2X7R antagonists, different animal models, and protocols, including a collection of samples throughout the study, to evaluate the dynamic changes that might occur in the parameters analyzed, particularly in the microbiota. Finally, antibiotics could provide additional relevant information on the exact role of the microbiota in the model and possibly confirm its potential synergism with the ATP/P2X7R pathway for the activation of the inflammasome and the full development of colitis and CA-CRC.

## 4. Materials and Methods

### 4.1. Ethics Statement

The ethics statement regarding animal experiments is provided in the Appendix A.

### 4.2. Animal Model and P2X7R-Blockade

Age-matched male 6-week-old (18–20 g) C57BL/6 P2X7^+/+^ and C57BL/6 P2X7^−^^/^^−^ mice (originally from The Jackson Laboratory, Bar Harbor, ME, USA) were maintained under specific pathogen-free conditions on a 14-h/10-h light and dark cycle in a temperature-controlled room (20–25 °C) with a relative humidity of 44–55%.

After an acclimation period of 1 week, mice were randomly assigned to 1 of 5 groups of 3–5 animals each (in 5 experiments). CA-CRC was induced with a single intraperitoneal (i.p.) injection of 12.5 mg/kg azoxymethane (AOM) (Sigma Aldrich, St. Louis, MO, USA). After seven days, the animals were given drinking water containing 2.5% dextran sodium sulfate (DSS) salt, 40–50 kDa (Spectrum, Gardena, CA, USA), for seven days in three cycles, with two intervals of 14 days each (between cycles 1 and 2 and between cycles 2 and 3) with drinking water without DSS. In a therapeutic protocol, P2X7R^+/+^ mice were treated with intraperitoneal injections of A-740003 (Tocris Bioscience, Bristol, UK), a P2X7R-selective antagonist, 1 h prior to the second and third cycles of DSS. The mice were weighed weekly and euthanized between Days 56 and 57 by inhalation of carbon dioxide (CO_2_). Additional details on the care of animals and study protocols are presented in Appendix A [64,65,66].

### 4.3. Colonoscopy Assessment

The details regarding the colonoscopy assessment are provided in Appendix A [20,67].

### 4.4. Histological Analysis

The distal 2 cm of the colon was divided into portions to perform all the experimental procedures. Colon samples were fixed in 40 g/L formaldehyde saline, embedded in paraffin, and 5 μm sections were then stained with H&E, periodic acid of Schiff (PAS), and phosphomolybdic acid picrosirius red dye and examined microscopically by two independent observers. Details regarding the histologic assessment are provided in Appendix A [33,68].

### 4.5. Immunohistochemistry and Assessment of Apoptosis

The details regarding immunohistochemistry and evaluation of apoptosis are provided in the Appendix A.

### 4.6. Quantitative Assessment of Colon Sections

The details of the quantitative analysis of tissue sections are described in Appendix A.

### 4.7. Assessment of Cytokines

Cytokine concentrations in 24 h culture supernatants of colon explants were measured via flow cytometry using the BD™ Cytometric Bead Array (CBA) Mouse Th1/Th2/Th17 Cytokine Kit (BD Biosciences, San Jose, CA, USA).

### 4.8. Analysis of Messenger RNA Expression

Gene expression was assessed by real-time reverse-transcription (RT) polymerase chain reaction (PCR). The procedures are described in the Appendix A.

### 4.9. Methylation Studies

The details regarding the methylation studies are described in Appendix A [69].

### 4.10. Microbiome Composition

Details regarding the sequencing of the variable V3–V4 regions of the 16S rRNA gene are described in Appendix A [70,71,72,73,74,75,76,77].

### 4.11. Statistical Analysis

The details regarding the statistical analysis are provided in the Appendix A.

## 5. Conclusions

Taken together, these findings support the participation of the ATP-P2X7R pathway in establishing an inflammatory microenvironment favoring the development of CA-CRC. These regulatory mechanisms activated downstream of P2X7R, in combination with signals from a dysbiotic microbiota, probably contribute to the maintenance and amplification of the inflammatory response resulting from the crosstalk of converging intracellular signaling pathways and the inflammasome. In addition, these mechanisms conveyed by P2X7R activation may be involved in the disruption of immune surveillance against tumor cells. Thus, we speculate that targeting P2X7R in IBD may constitute a potential new approach not only for preventing and treating inflammation but also for preventing and even contributing to the treatment of CA-CRC.

## Figures and Tables

**Figure 1 ijms-23-04616-f001:**
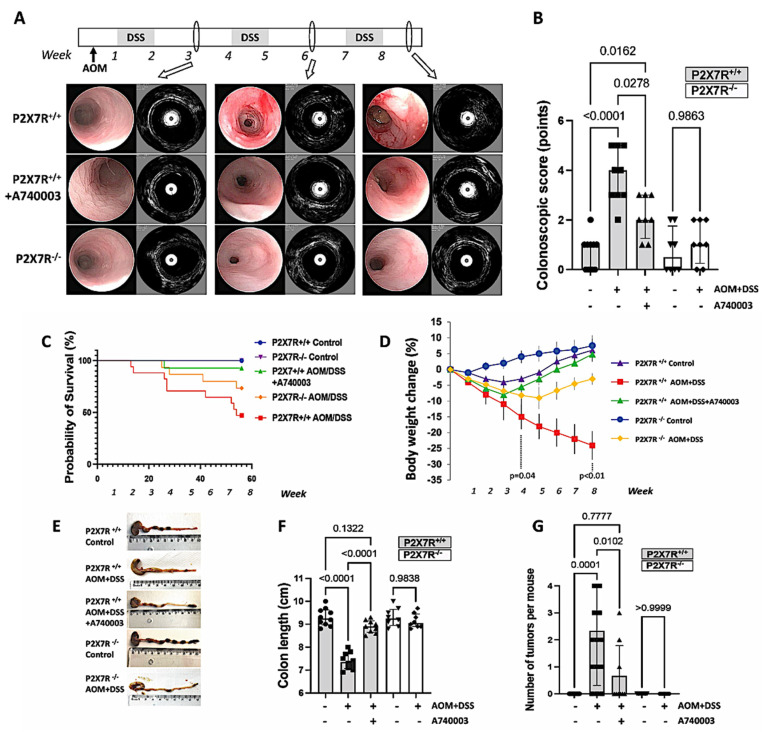
P2X7R^−^^/^^−^ and P2X7R^+/+^ mice treated with the P2X7R antagonist A740003 do not develop full AOM/DSS-induced colitis or CA-CRC. Video colonoscopy associated with 3-D endoluminal ultrasound biomicroscopy imaging was performed after induction at weeks 3, 6, and 8 (**A**). Values are medians with interquartile ranges of 8–10 animals per group. Significant values are presented (**B**). The survival curves for the AOM/DSS-induced P2X7R^+/+^ mice showed a significant increase in mortality compared with that of the other groups at week 8 (**C**). The AOM/DSS-induced P2X7R^+/+^ mice presented progressive weight loss compared with the mice in the other groups (**D**); reduced colon length (**E**,**F**); and a smaller number of polyps/tumors per animal (**G**). The survival curves were analyzed, and the *p* values were determined by the log-rank test. Values are medians with interquartile ranges of three independent experiments, with 5–7 animals per group. Significant values are presented.

**Figure 2 ijms-23-04616-f002:**
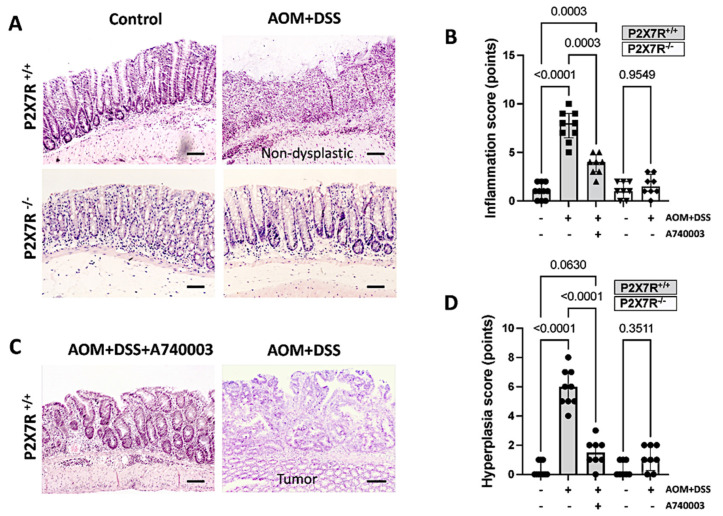
P2X7R blockade attenuates colonic injury in the AOM/DSS-treated mice. Histopathological analysis by hematoxylin and eosin (HE) staining of the colon shows the typical inflammatory changes and tissue damage induced by AOM/DSS in P2X7R^+/+^ mice (**A**). The P2X7R^−^^/^^−^ and P2X7R^+/+^ mice treated with the P2X7R antagonist A740003 developed significantly fewer inflammatory changes (**B**) and tumors (**C**,**D**). Values are medians with interquartile ranges of three independent experiments, with 4–5 animals per group. The scale bars represent 20 μm. The analysis was performed by Brown-Forsythe and Welch ANOVA tests, in which multiple comparisons were carried out using Dunnett’s T3 test. Significant values are presented.

**Figure 3 ijms-23-04616-f003:**
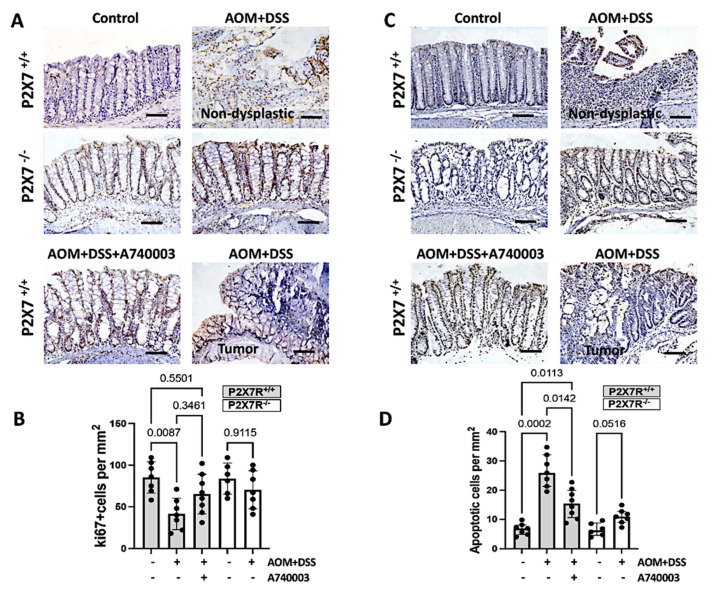
P2X7R blockade protects against colonic injury and apoptotic cell loss in the AOM/DSS-treated mice. Proliferating cells were immunohistochemically labeled with anti-Ki67 antibody (**A**), and apoptotic cells were detected using a TUNEL assay (**C**), as shown by the representative photomicrographs. In nondysplastic inflamed areas, the P2X7R^−^^/^^−^ and P2X7R^+/+^ mice treated with the P2X7R antagonist A740003 did not exhibit a change in the rate of Ki67-positive cells (**B**) but developed significantly less apoptosis (**D**). In the tumor areas of the P2X7R^+/+^ treated mice, the number of Ki67-positive cells was increased, while the number of apoptotic cells was decreased. Values are medians with interquartile ranges of three independent experiments, with 4–5 animals per group. The scale bars represent 20 μm. Significant values are presented. The analysis was performed by Brown-Forsythe and Welch ANOVA tests, in which multiple comparisons were carried out using Dunnett’s T3 test. Significant values are presented.

**Figure 4 ijms-23-04616-f004:**
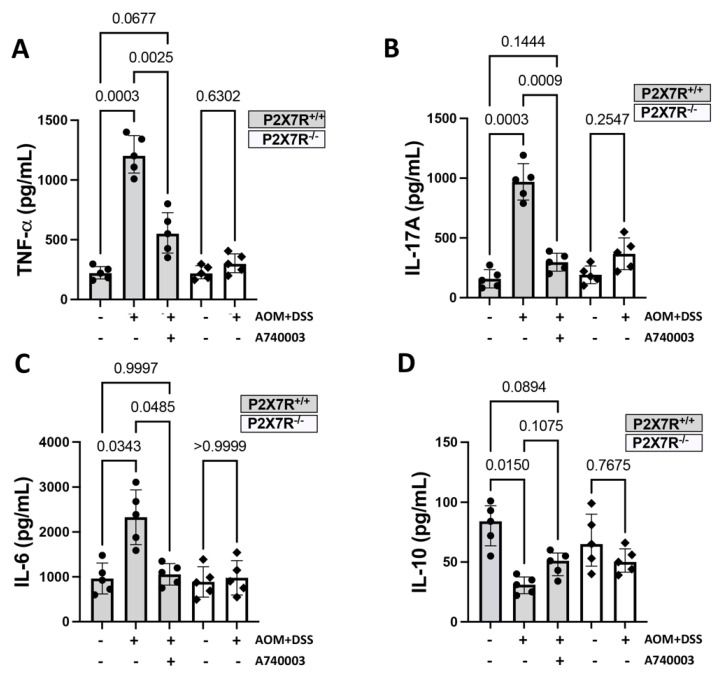
P2X7R blockade modulates proinflammatory cytokine production in colon explants from the AOM/DSS-treated mice. Supernatants from colon explants cultured for 24 h at 37 °C with 5% CO_2_ were used to measure the concentrations of cytokines by CBA. The P2X7R^−^^/^^−^ and P2X7R^+/+^ mice treated with the P2X7R antagonist A740003 developed significantly lower concentrations of TNF-alpha (**A**), IL-17A (**B**), and IL-6 (**C**) and higher concentrations of IL-10 than the P2X7R^+/+^ controls (**D**). The results are expressed as pg/mL. Values are medians with interquartile ranges of three independent experiments, with 4–5 animals per group. Significant values are presented. The analysis was performed by Brown-Forsythe and Welch ANOVA tests, in which multiple comparisons were carried out using Dunnett’s T3 test. Significant values are presented.

**Figure 5 ijms-23-04616-f005:**
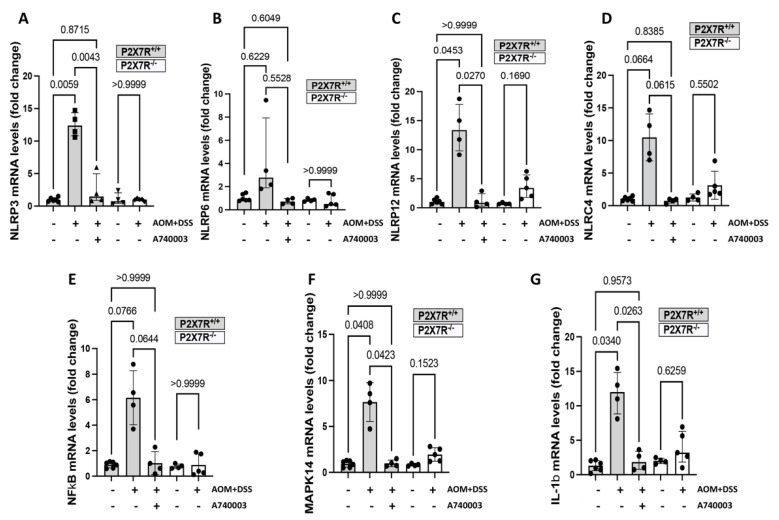
P2X7R blockade modulates the expression of genes related to inflammation and cancer in the colon. The mRNA levels measured by quantitative real-time PCR in colon samples of the P2X7R^−^^/^^−^ and P2X7R^+/+^ mice treated with the P2X7R antagonist A740003 were significantly lower than those in the P2X7R^+/+^-treated animals for the *NLRP3* (**A**), *NLRP12* (**C**), *MAPK14* (**F**), and *IL-1beta* (**G**) genes. Other genes analyzed in this study did not show significant changes among the experimental groups (**B**,**D**,**E**). Values are medians with interquartile ranges of three independent experiments, with 3–4 animals per group. The analysis was performed by Brown-Forsythe and Welch ANOVA tests, in which multiple comparisons were carried out using Dunnett’s T3 test. Significant values are presented.

**Figure 6 ijms-23-04616-f006:**
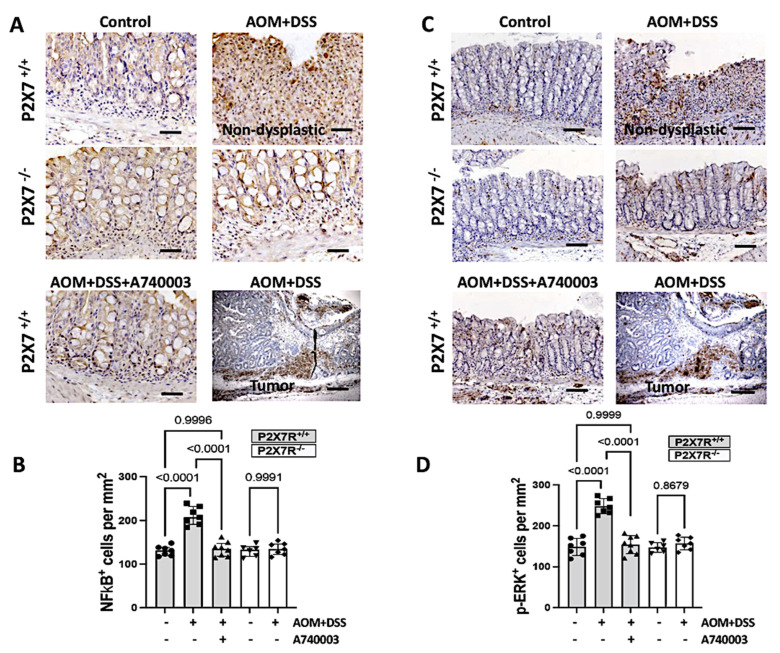
P2X7R blockade attenuates the expression of intracellular signaling pathways involved in cytokine production and cell survival triggered by AOM/DSS induction. The P2X7R^−^^/^^−^ and P2X7R^+/+^ mice treated with the P2X7R antagonist A740003 expressed less NF-kappa B (**A**,**B**) and phospho-ERK (**C**,**D**) than the P2X7R^+/+^-treated animals. Values are medians with interquartile ranges of three independent experiments, with 4–5 animals per group. The scale bars represent 20 μm. The analysis was performed by Brown-Forsythe and Welch ANOVA tests, in which multiple comparisons were carried out using Dunnett’s T3 test. Significant values are presented.

**Figure 7 ijms-23-04616-f007:**
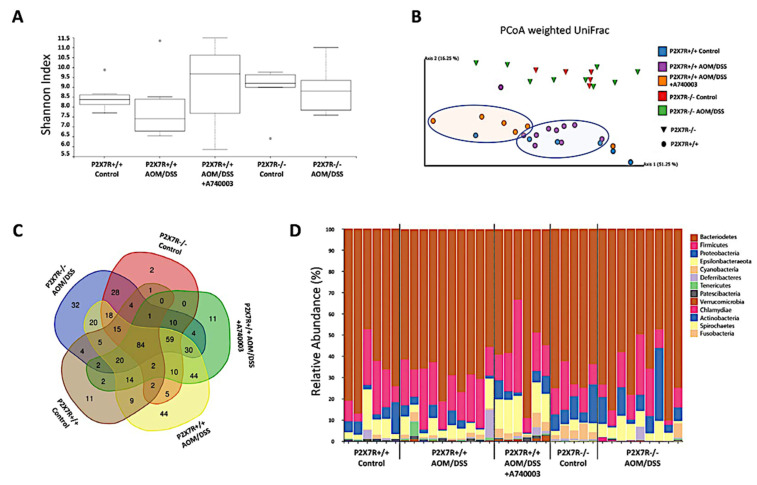
P2X7R blockade is followed by changes in the fecal microbial diversity and composition. Alpha-diversity analysis using the Shannon index shows an upward trend in fecal microbial diversity when P2X7R is blocked (**A**). Beta-diversity analysis using weighted UniFrac principal coordinate analysis (PCoA) clusters P2X7R^−/−^ away from P2X7R^+/+^ samples (**B**). A Venn diagram displays the logical relations among groups (**C**). Differential abundance analysis of taxonomic profiles depicts the microbial composition at the phylum level (**D**). Sequencing was performed with samples from two independent experiments, with 2–5 animals per group.

## Data Availability

Materials, such as protocols, analytic methods, and study material, are available upon request to interested researchers. The raw data supporting the conclusions of this manuscript will be made available by the authors, without undue reservation, to any qualified researcher. Data regarding the microbiota sequencing are available through the accession numbers: Submission ID: SUB11172419; BioProject: ID: PRJNA814660.

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
