# Peer review of "The P2X7 Receptor Promotes Colorectal Inflammation and Tumorigenesis by Modulating Gut Microbiota and the Inflammasome"

_ijms, 2022, doi:10.3390/ijms23094616_

Round 1
Reviewer 1 Report
This manuscript is describing about the promotion of colorectal inflammation and tumorigenesis by P2X7 receptor via modulating gut microbiota and the inflammasome. The authors studied the role of P2X7 receptor in the development and progression of colitis-associated colorectal cancer (CA-CRC) using biomicroscopy and colon tissue analyses for P2X7R+/+ and P2X7R-/- mice. They found that these regulatory mechanisms activated downstream of P2X7R, in combination with signals from a dysbiotic microbiota, probably via the inflammatory response resulting from the crosstalk of converging intracellular signaling pathways and the inflammasome. It is recommended to be acceptable after minor revision.
- The last sentence in abstract should be rewritten more clearly.
- In Figure 2, letter B in left side -> be deleted. D -> C
- "B" should be added in legend of Figure 3.
- Panel labels or legend in Figure 6 should be revised.
Author Response
Response to reviewers:
On behalf of my group, I would like to thank all the reviewers for their attentive and
detailed evaluation of the manuscript, and for the opportunity to improve the manuscript based
on their insightful comments and suggestions.
We believe that we have addressed all the reviewers’ concerns and comments. To this
end, several sections of the manuscript were revised, thereby improving reader's understanding.
Modifications suggested by the reviewers have polished the manuscript and increased its overall
impact.
All revisions to the manuscript have been marked up using the “Track Changes” function
of MS Word.
Reviewer Comments
Reviewer #1
This manuscript is describing about the promotion of colorectal inflammation and tumorigenesis by P2X7
receptor via modulating gut microbiota and the inflammasome. The authors studied the role of P2X7
receptor in the development and progression of colitis-associated colorectal cancer (CA-CRC) using
biomicroscopy and colon tissue analyses for P2X7R+/+ and P2X7R-/- mice. They found that these
regulatory mechanisms activated downstream of P2X7R, in combination with signals from a dysbiotic
microbiota, probably via the inflammatory response resulting from the crosstalk of converging
intracellular signaling pathways and the inflammasome. It is recommended to be acceptable after minor
revision.
R: We greatly appreciate this reviewer’s helpful comments and thank him/her for understanding
the importance of our work and for supporting our study.
• The last sentence in abstract should be rewritten more clearly.
R: We attempted to clarify the meaning of the last sentence in the Abstract, as suggested.
3
• In Figure 2, letter B in left side -> be deleted. D -> C
R: This reviewer is correct. We made the appropriate changes as suggested by this reviewer.
• "B" should be added in legend of Figure 3.
R: We made the appropriate changes as suggested by this reviewer.
• Panel labels or legend in Figure 6 should be revised.
R: We made the appropriate revision as suggested by this reviewer.
Reviewer 2 Report
There is a consistent literature on this well-known purinergic receptor and its expanding implications in inflammation and tumorigenesis. Authors are invited to enrich their references list also to better highlight to the readers its anti- and also pro-tumorigenesis effect and intimate relation with intracellular ATP stimulation modalitues. Comparative elaboration ves the few other inhibitors would be good foe the readers in view, if possible, to point out the one used in this study. An permeability work ou would have been an added value. Authors should pay attention to inadvertent plagiarism by providing a validated check.
Author Response
Response to reviewers:
On behalf of my group, I would like to thank all the reviewers for their attentive and
detailed evaluation of the manuscript, and for the opportunity to improve the manuscript based
on their insightful comments and suggestions.
We believe that we have addressed all the reviewers’ concerns and comments. To this
end, several sections of the manuscript were revised, thereby improving reader's understanding.
Modifications suggested by the reviewers have polished the manuscript and increased its overall
impact.
All revisions to the manuscript have been marked up using the “Track Changes” function
of MS Word.
Reviewer Comments
Reviewer #2
There is a consistent literature on this well-known purinergic receptor and its expanding implications in
inflammation and tumorigenesis. Authors are invited to enrich their references list also to better highlight
to the readers its anti- and also pro-tumorigenesis effect and intimate relation with intracellular ATP
stimulation modalitues.
R: We greatly appreciate this reviewer’s helpful comments and thank him/her for understanding
the importance of our work and for supporting our study. We agree with this reviewer’s
suggestion of including additional references on the subject. We added 5 new references to the
Introduction section, and 10 to the Discussion section.
New references included:
Introduction:
13 Lammas DA, Stober C, Harvey CJ, Kendrick N, Panchalingam S, Kumararatne DS. ATP-induced
killing of mycobacteria by human macrophages is mediated by purinergic P2Z(P2X7) receptors. Immunity
1997; 7(3): 433-444 [PMID: 9324363 DOI: 10.1016/s1074-7613(00)80364-7]
3
14 Fairbairn IP, Stober CB, Kumararatne DS, Lammas DA. ATP-mediated killing of intracellular
mycobacteria by macrophages is a P2X(7)-dependent process inducing bacterial death by phagosome-
lysosome fusion. J Immunol 2001; 167(6): 3300-3307 [PMID: 11544318 DOI: 10.4049/jimmunol.167.6.3300]
16 Pelegrin P, Surprenant A. Pannexin-1 mediates large pore formation and interleukin-1beta release
by the ATP-gated P2X7 receptor. EMBO J 2006; 25(21): 5071-5082 [PMID: 17036048 PMCID: PMC1630421
DOI: 10.1038/sj.emboj.7601378]
17 Kawamura H, Aswad F, Minagawa M, Govindarajan S, Dennert G. P2X7 receptors regulate NKT
cells in autoimmune hepatitis. J Immunol 2006; 176(4): 2152-2160 [PMID: 16455971 DOI:
10.4049/jimmunol.176.4.2152]
18 Chen YG, Scheuplein F, Driver JP, Hewes AA, Reifsnyder PC, Leiter EH, Serreze DV. Testing the
role of P2X7 receptors in the development of type 1 diabetes in nonobese diabetic mice. J Immunol 2011;
186(7): 4278-4284 [PMID: 21357538 PMCID: PMC3094905 DOI: 10.4049/jimmunol.1003733]
Discussion:
24 Adinolfi E, Capece M, Franceschini A, Falzoni S, Giuliani AL, Rotondo A, Sarti AC, Bonora M,
Syberg S, Corigliano D, Pinton P, Jorgensen NR, Abelli L, Emionite L, Raffaghello L, Pistoia V, Di Virgilio
F. Accelerated tumor progression in mice lacking the ATP receptor P2X7. Cancer Res 2015; 75(4): 635-644
[PMID: 25542861 DOI: 10.1158/0008-5472.CAN-14-1259]
25 Ghiringhelli F, Apetoh L, Tesniere A, Aymeric L, Ma Y, Ortiz C, Vermaelen K, Panaretakis T,
Mignot G, Ullrich E, Perfettini JL, Schlemmer F, Tasdemir E, Uhl M, Genin P, Civas A, Ryffel B,
Kanellopoulos J, Tschopp J, Andre F, Lidereau R, McLaughlin NM, Haynes NM, Smyth MJ, Kroemer G,
Zitvogel L. Activation of the NLRP3 inflammasome in dendritic cells induces IL-1beta-dependent adaptive
immunity against tumors. Nat Med 2009; 15(10): 1170-1178 [PMID: 19767732 DOI: 10.1038/nm.2028]
26 Sougiannis AT, VanderVeen B, Chatzistamou I, Kubinak JL, Nagarkatti M, Fan D, Murphy EA.
Emodin reduces tumor burden by diminishing M2-like macrophages in colorectal cancer. Am J Physiol
Gastrointest Liver Physiol 2022; 322(3): G383-G395 [PMID: 35018819 PMCID: PMC8897011 DOI:
10.1152/ajpgi.00303.2021]
27 Zhang Y, Pu W, Bousquenaud M, Cattin S, Zaric J, Sun LK, Ruegg C. Emodin Inhibits
Inflammation, Carcinogenesis, and Cancer Progression in the AOM/DSS Model of Colitis-Associated
Intestinal Tumorigenesis. Front Oncol 2020; 10: 564674 [PMID: 33489875 PMCID: PMC7821392 DOI:
10.3389/fonc.2020.564674]
29 Zhang Y, Li F, Wang L, Lou Y. A438079 affects colorectal cancer cell proliferation, migration,
apoptosis, and pyroptosis by inhibiting the P2X7 receptor. Biochem Biophys Res Commun 2021; 558: 147-153
[PMID: 33915328 DOI: 10.1016/j.bbrc.2021.04.076]
33 Jin BR, Chung KS, Hwang S, Hwang SN, Rhee KJ, Lee M, An HJ. Rosmarinic acid represses colitis-
associated colon cancer: A pivotal involvement of the TLR4-mediated NF-kappaB-STAT3 axis. Neoplasia
2021; 23(6): 561-573 [PMID: 34077834 PMCID: PMC8180929 DOI: 10.1016/j.neo.2021.05.002]
37 Wan P, Liu X, Xiong Y, Ren Y, Chen J, Lu N, Guo Y, Bai A. Extracellular ATP mediates
inflammatory responses in colitis via P2 x 7 receptor signaling. Sci Rep 2016; 6: 19108 [PMID: 26739809
PMCID: PMC4703960 DOI: 10.1038/srep19108]
41 Landskron G, De la Fuente M, Thuwajit P, Thuwajit C, Hermoso MA. Chronic inflammation and
cytokines in the tumor microenvironment. J Immunol Res 2014; 2014: 149185 [PMID: 24901008 PMCID:
PMC4036716 DOI: 10.1155/2014/149185]
4
43 Luo C, Zhang H. The Role of Proinflammatory Pathways in the Pathogenesis of Colitis-Associated
Colorectal Cancer. Mediators Inflamm 2017; 2017: 5126048 [PMID: 28852270 PMCID: PMC5568615 DOI:
10.1155/2017/5126048]
47 Shannon S, Jia D, Entersz I, Beelen P, Yu M, Carcione C, Carcione J, Mahtabfar A, Vaca C, Weaver
M, Shreiber D, Zahn JD, Liu L, Lin H, Foty RA. Inhibition of glioblastoma dispersal by the MEK inhibitor
PD0325901. BMC Cancer 2017; 17(1): 121 [PMID: 28187762 PMCID: PMC5303286 DOI: 10.1186/s12885-017-
3107-x]
Comparative elaboration ves the few other inhibitors would be good foe the readers in view, if possible,
to point out the one used in this study.
R: We understand this reviewer’s point of view and we added some new text to the manuscript
(please see the new references 26, 27, and 29).
An permeability work ou would have been an added value.
R: We agree with this reviewer’s concern and expectations. However, the study was already
too long with many different experiments to show and interpret. Actually, we are planning to
add permeability experiments in future studies.
Authors should pay attention to inadvertent plagiarism by providing a validated check.
R: We understand this reviewer’s concerns. For such purpose, we used the Grammarly
plagiarism check tool, which identified 4% “plagiarism” in the submitted version, before any
changes have been performed. The information button indicated that 4% of our text matches 20
fragments from 9 sources on the web or in academic databases.
The indicated fragments frequently came from our own previous studies, reflecting similarities
in the settings regarding Material and Methods and the Results sections. The contents of the
fragments are basically non-specific, referring to details on methods, statistical analysis,
description of results, and figure legends. Please, see the fragments captured by Grammarly
below.
Highlighted fragments identified by similarity (order of appearance):
1) ... as a damage-associated molecular pattern (DAMP) capable of...
2) ... Therefore, in the present study, we investigated the...
3) ... The analysis was performed by Brown-Forsythe and Welch ANOVA tests, in which
multiple comparisons were....
4) ... The analysis was performed by Brown-Forsythe and Welch ANOVA tests, in which
multiple comparisons were....
5) ... The analysis was performed by Brown-Forsythe and Welch ANOVA tests, in which
multiple comparisons were....
6) ... The analysis was performed by Brown-Forsythe and Welch ANOVA tests, in which
multiple comparisons were....
5
7) ... The analysis was performed by Brown-Forsythe and Welch ANOVA tests, in which
multiple comparisons were....
8) ... The analysis was performed by Brown-Forsythe and Welch ANOVA tests, in which
multiple comparisons were....
9) ...while the number of apoptotic cells was decreased.
10) ... To characterize the different cell populations present in the...
11) ... The dense inflammatory cell infiltration observed in the lamina propria of...
12) ... Here, we present only the quantifiable results obtained. The analysis of the
supernatants obtained from...
13) ... were used to measure the concentrations of cytokines by CBA....
14) ... displayed similar expression patterns and tissue distributions and were present in
both the epithelium and the lamina propria mononuclear cells at significantly higher
densities in...
15) ... The Firmicutes:Bacteroidetes ratio was not significantly different...
16) ... To our knowledge, only one study has investigated the...
17) ... This finding appears to be in accordance with...
18) ... serve, at least in part, as a guide for future investigations. Further studies on the
subject should also consider other dosages...
19) ...After an acclimation period of 1 week, mice were randomly
20) ... the experimental procedures. Colon samples were fixed in 40 g/L formaldehyde
saline, embedded in paraffin,...